# Joint Liver Lesion Segmentation and Classification via Transfer Learning

**Michal Heker**[1]                                            MICHALHEKER@GMAIL.COM
**Hayit Greenspan**[1]                                         HAYIT@ENG.TAU.AC.IL
[1] *Department of Biomedical Engineering, Tel-Aviv University, Israel*

## Abstract

Transfer learning and joint learning approaches are extensively used to improve the performance of Convolutional Neural Networks (CNNs). In medical imaging applications in which the target dataset is typically very small, transfer learning improves feature learning while joint learning has shown effectiveness in improving the network's generalization and robustness. In this work, we study the combination of these two approaches for the problem of liver lesion segmentation and classification. For this purpose, 332 abdominal CT slices containing lesion segmentation and classification of three lesion types are evaluated. For feature learning, the dataset of MICCAI 2017 Liver Tumor Segmentation (LiTS) Challenge is used. Joint learning shows improvement in both segmentation and classification results. We show that a simple joint framework outperforms the commonly used multi-task architecture (Y-Net), achieving an improvement of 10% in classification accuracy, compared to a 3% improvement with Y-Net.

**Keywords:** joint learning, liver lesions, lesion classification, lesion segmentation, CT

## 1. Introduction

Deep learning methodologies, especially Convolutional Neural Networks (CNNs), are the top performers in most medical image processing tasks, including liver lesion segmentation and classification in abdominal CT images (Litjens et al., 2017),(Ben-Cohen et al., 2016),(Heker et al., 2019). Automatic segmentation and classification is challenging due to different liver lesions contrast, and considerable visual appearance variability between lesion types.

Lesion segmentation has attracted attention in recent years, with publicly available datasets that enable comparison between different methods (Bilic et al., 2019),(Soler et al., 2010). Lesion classification, on the other hand, is far less investigated with very limited-size datasets explored and no public data available. Transfer learning and joint training are two of the approaches used to address the limited data challenge and improve the performance of CNNs. Joint learning, including multi-task learning, has been shown to improve network generalization, resulting in better performance on a given target task (Ruder, 2017).

In this work, we compare different approaches to transfer learning with the goal of improving joint segmentation and classification of liver lesions. We introduce and compare the results for two U-Net based frameworks that incorporate joint learning. The following contributions are included in this research: (1) We focus on transfer learning with the use of data from a similar domain and related target task. (2) Two U-Net based frameworks

that combine transfer learning and joint learning of segmentation and classification are introduced and evaluated.

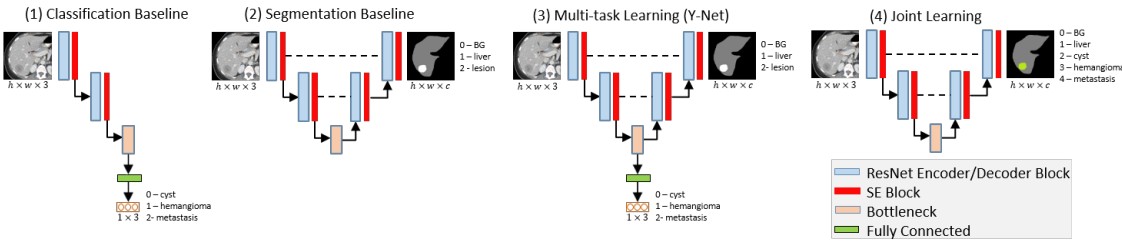

Figure 1: The proposed frameworks: (1) SE-Resnet for classification. (2) U-Net with SE-Resnet backbone for segmentation. (3) Y-Net like architecture, i.e. U-Net with SE-Resnet backbone with 2 separate output branches; a classification output at the end of the encoder and a segmentation output at the end of the decoder. (4) U-Net with SE-Resnet backbone for segmentation by pixel-wise classification.

## 2. Proposed Frameworks for Joint Segmentation and Classification

The proposed frameworks are based on a U-Net model with a SE-ResNet encoder, combining residual skip connections and Squeeze-and-Excitation blocks (SE blocks) (Ronneberger et al., 2015), (He et al., 2016), (Hu et al., 2018). The U-Net architecture enables the encoder and the decoder to share information and learn both global and local features to produce quality segmentation. The first framework incorporates a multi-task U-Net, similarly to (Mehta et al., 2018),(Le et al., 2019), where joint semantic segmentation and global classification are separated into two distinct outputs and trained simultaneously. The segmentation output is extracted from the U-net's decoder as in a vanilla U-Net, while the classification output is extracted as a parallel branch from the U-Net's encoder. We use a multi-class loss $L = \lambda L_{seg} + (1 - \lambda)L_{cls}$, where weighted cross-entropy loss is used for segmentation to balance the classes (the lesion class is under-represented compared to the liver and background classes), and categorical cross-entropy is used for classification. In the second framework, segmentation and classification are combined into one output that performs semantic segmentation (pixel-wise classification), where the final lesion class is determined using a majority vote. Weighted cross-entropy loss is used here as well, with suitable weights for each class, reversely proportional to their ratio in the dataset. Additionally, classification and segmentation models were trained individually to get baseline performances on each task separately. The different frameworks are illustrated in Figure 1.

We train and evaluate the models on 332 2D CT slices from Sheba Medical Center (Israel) with medical records from 140 patients for cases of 3 types of liver lesions with the following distribution: 75 cysts, 71 hemangiomas, 93 metastases. Since the dataset size is small, we use 3-fold cross-validation to evaluate the performance of the models.

We explore the benefit of using transfer learning by fine-tuning the frameworks in different approaches: (1) training from scratch (random initialization), (2) fine-tuning ImageNet

weights (Russakovsky et al., 2015), (3) fine-tuning the weights of a self-trained lesion segmentation model, trained on the dataset of MICCAI 2017 Liver Tumor Segmentation (LiTS) Challenge. The LiTS dataset contains thousands of lesion images, which enables training a strong and robust lesion segmentation model (Bilic et al., 2019).

It is worth mentioning that we use liver crops as input during training and not the entire CT slice, which helps the class-imbalance issue. This is done by using a 2-stage cascaded approach, where a first network is trained on the LiTS dataset to obtain high quality liver segmentation (achieving competitive results of 96.1% in Dice per case score on the LiTS challenge leader board). This network is used on our private dataset to extract liver ROI crops that are used as inputs for the trained models.

## 3. Results and Conclusions

Table 1 reports the segmentation and classification performance of the frameworks trained with different weights initialization. Dice coefficient, Recall, and classification accuracy are used for evaluation. Qualitative results are presented in Figure 2.

Table 1: Performance comparison of segmentation and classification

| Training strategy | Fine-tuning | Cls Acc | Seg Dice | Seg Recall |
|---|---|---|---|---|
| 1. Classification baseline | **scratch** | 0.55 | - | - |
| | **ImageNet** | 0.63 | - | - |
| | **LiTS** | 0.76 | - | - |
| 2. Segmentation baseline | **scratch** | - | 0.59 | 0.59 |
| | **ImageNet** | - | 0.63 | 0.67 |
| | **LiTS** | - | 0.71 | 0.72 |
| 3. Multi-task learning (Y-Net) | **scratch** | 0.43 | 0.49 | 0.43 |
| | **ImageNet** | 0.68 | 0.67 | 0.65 |
| | **LiTS** | 0.79 | 0.71 | 0.68 |
| 4. Joint learning | **scratch** | 0.63 | 0.57 | 0.60 |
| | **ImageNet** | 0.74 | 0.64 | 0.70 |
| | **LiTS** | **0.86** | **0.71** | **0.76** |

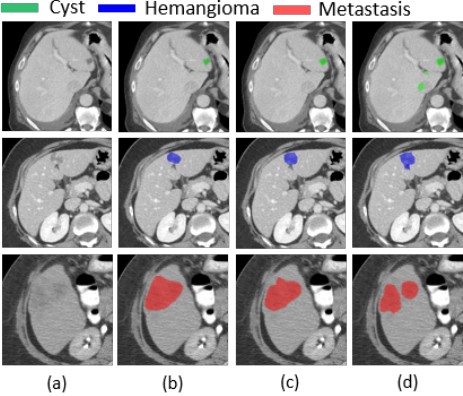

Figure 2: Lesion segmentation and classification results: (a) input image; (b) ground truth; (c) joint learning; (d) multi-task Y-Net.

We observe that the joint learning frameworks (see Table 1: 3,4) outperform individual task learning. Interestingly, the simple joint framework achieved the best results in both segmentation and classification tasks, outperforming the commonly used multi-task architecture (Y-Net). Compared to the baseline models, Y-net showed 3% improvement in classification accuracy. The joint learning framework showed large improvements: 10% in classification and 4% in segmentation recall with the same dice score.

Using transfer learning generally improved weight initialization and resulted in faster convergence. ImageNet pre-training showed improved convergence and accuracy. Pre-training with LiTS enabled the model to learn shared representations in similar domains, resulting in better generalization and higher accuracy.

From the evaluation study conducted, we conclude that in the joint network classification and localization context are shared for mutual benefit, thus increasing results. A second conclusion is that transferring learned models via pre-training of networks between two similar tasks provides for stronger and robust representations. In future work, we hope to show the generalization to other domains and tasks.

## Acknowledgments

This research was supported by the Israel Science Foundation (grant No. 1918/16).

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
