# OpenReview forum: "Joint Liver Lesion Segmentation and Classification via Transfer Learning"
_MIDL.io/2020/Conference — MIDL 2020_

### Official Review · AnonReviewer4 · 2020-03-02
**Some details missing, but looks like good work**

**Rating:** 4
**Confidence:** 5

**Review:**

This work is based on a private dataset of 332 CT slices (not volumes!) from 140 patients with three different types of annotated liver lesions (cysts, hemangiomas, mets). They compare two multi-task approaches for segmentation and classification and two baseline (ablation, single-task) approaches on this dataset. Additionally, the encoders are either randomly initialised or pretrained on ImageNet (out of domain) or the LiTS dataset (same domain).

The number of 2D slices being used is relatively small, which limits the contribution to some degree, but the setup is solid and the results definitely interesting.

I missed some details on the architectures (numbers of filters, for instance) and possible image preprocessing.  I also wondered if the number of resolution levels is really only 3, which would limit the receptive field (without knowing any details about the employed blocks, it is hard to guess, but it could be around 44² pixels theoretical maximum, the ERF being even smaller).

It would also have been interesting to do an ablation study on the SE (squeeze & excitation) blocks, but at least they were used in all four compared approaches, so the comparison is fair.

Overall, I would rate it between 3 and 4, but I do think it is a nice contribution to MIDL, so I voted 4 ("strong" accept).

---

### Official Review · AnonReviewer3 · 2020-03-10
**no novel approach; missing paper focus on application**

**Rating:** 2
**Confidence:** 3

**Review:**

(+) An exhaustive validation of different experimental setups for liver lesion segmentation and classification on a self-collected database is provided.

(+) Figure 1 nicely summarizes the tasks and experimental setups.

(+/-) The paper is mostly well written. However, I would recomment to restructure Section 2. Your paper is of type well-validated application. Thus, first describing the given segmentation and classification tasks and the collected data and afterwards experimental setups seems more resonable to me. In general, the paper focuses too much on the methodology of transfer and joint learning to my opinion.

(-) Your motivation is quite weak. Why is such an automatic classification approach required? Please specify in abstract/introduction.

(-) No comparison to existing approaches on liver lesion segmentation (e.g. winners of the LiTS Challenge) is performed.

(-) Transfer, joint and multi-task learning are well known approaches to deal with limited data. No methodical tricks are presented.

(-) "The  first  framework  incorporates  a  multi-task  U-Net, [...]" This is confusing, as Figure 1 shows the multi-task approach as third framework.

---

### Official Review · AnonReviewer2 · 2020-03-10
**Good application of previous techniques**

**Rating:** 3
**Confidence:** 5

**Review:**

In this paper, the authors combine the advantages of joint learning and transfer learning to improve the performance on
liver lesion segmentation and classification. Although the techniques are not new, it is good to use them in new applications.

I am just curious about the performance in the following settings:
(1) The segmentation performance on the authors' private data using the pretrained model from LiTS dataset.
(2) The segmentation performance on the authors' private data if only finetuning a segmentation model instead of the joint model.

Setting (1) can show us how much performance the finetuning can improve based on the good pretrained model from LiTS dataset.
Setting (2) can show us whether the joint learning has a bad effect on the segmentation task.

---

### Official Review · AnonReviewer1 · 2020-03-13
**Good short paper, requires some clarifications**

**Rating:** 3
**Confidence:** 4

**Review:**

I found the paper easy to read and containing interesting results about how some baseline models ourperform more sophisticated ones. I am recommending acceptance, but I have some remaining doubts that I would like to be answered, if possible. Namely:
1) I do not see very clearly from the text what the authors mean by joint learning. I believe Figure 1 could serve the purpose of actually clarifying what is happening in each scenario; unfortunately, it has a very poor caption. Could the authors add a short description of each of the four schemes in that caption, and label them as a), b), c), and d)? I believe that would help a lot.
2) I don't understand this sentence "the lesion class is under-represented". I guess it is because of the use of the word "class", which makes the reader think about classification. Do you actually mean "slices containing liver lesions were under-represented as opposed as lesion-free slices"? Because later in the text, the number of examples for each lesion class is mentioned, and they seem pretty balanced. Anyway, if what I am saying is the case, are you using weighted cross-entropy, rather you are oversampling slices with lesions during training?

---

### Meta-Review · Area_Chair1 · 2020-04-07
**MetaReview of Paper322 by AreaChair1**

**Rating:** 1

**Metareview:**

The following quotes from the reviews demonstrate important critical points sufficient to justify reject.
no rebuttal was provided to address any of them:

- "transfer, joint and multi-task learning are well known approaches to deal with limited data",... "the techniques are not new", "application of previous techniques", "no novel approach"

- "motivation is quite weak"

- "no comparison to existing approaches on liver lesion segmentation"


In summary, rejection is justified by lack of technical novelty, weak motivation, and lack of comparison. Most reviewers also pointed out some issues related to clarity and lack of details and focus.

**Paper Type:**

validation/application paper

---

### Decision · Program_Chairs · 2020-04-11

**Decision:**

Accept

**Comment:**

Taking all information into account, it was determined that the paper was accepted based on its merit.